# An operator view of policy gradient methods

**Dibya Ghosh**
Google Brain

**Marlos C. Machado**
Google Brain

**Nicolas Le Roux**
Google Brain

## Abstract

We cast policy gradient methods as the repeated application of two operators: a policy improvement operator $\mathcal{I}$, which maps any policy $\pi$ to a better one $\mathcal{I}\pi$, and a projection operator $\mathcal{P}$, which finds the best approximation of $\mathcal{I}\pi$ in the set of realizable policies. We use this framework to introduce operator-based versions of well-known policy gradient methods such as REINFORCE and PPO, which leads to a better understanding of their original counterparts. We also use the understanding we develop of the role of $\mathcal{I}$ and $\mathcal{P}$ to propose a new global lower bound of the expected return. This new perspective allows us to further bridge the gap between policy-based and value-based methods, showing how REINFORCE and the Bellman optimality operator, for example, can be seen as two sides of the same coin.

## 1 Introduction

Model-free reinforcement learning algorithms aim at learning a policy that maximizes the (discounted) sum of rewards directly from samples generated by the agent's interactions with the environment. These techniques mainly fall in one of two categories: value-based methods [e.g., 26, 15], where the agent predicts the value of taking an action and then chooses the action with the largest predicted value; and policy-based methods, where the agent directly learns a good distribution over actions at each state. Although several past works created connections between the two views [e.g., 11, 19], such connections are often limited to the optimal policy and they do not capture training dynamics.

In particular, value-based methods like fitted Q-iteration [15] and Q-learning [26] are often cast as the iterative application of an improvement operator, the Bellman optimality operator, which transforms the value function into a "better" one (unless the value function is already the optimal one). When dealing with a restricted set of policies, we often use function approximation for the value function. In this case, the learning procedure interleaves the improvement operator with a projection operator, which finds the best approximation of this improved value function in the space of realizable value functions.

While this view is the basis for many intuitions around the convergence of value-based methods, no such view exists for policy-gradient (PG) methods, which are usually cast as doing gradient ascent on a parametric function representing the expected return of the policy [e.g., 27, 18]. Although this property can be used to show the convergence when using a sufficiently small step-size, it does little to our understanding of the relationship of policy-gradient methods and value-based ones.

In this work, we show that PG methods can also be seen as repeatedly applying two operators akin to those encountered in value-based methods: (a) a policy improvement operator, which maps any policy to a policy achieving strictly larger return; and (b) a projection operator, which finds the best approximation of this new policy in the space of realizable policies. We then recast common PG methods under this framework, using their operator interpretations to shed light on their properties.

We also make the following additional contributions: (a) We present a lower bound on the performance of a policy using the state-action formulation, leading to an alternative to conservative policy improvement; (b) We provide a formal justification of $\alpha$-divergences in the imitation learning setting.

## 2 Background

We consider an infinite-horizon discounted Markov decision process (MDP) [14] defined by the tuple $\mathcal{M} = \langle \mathcal{S}, \mathcal{A}, p, r, d_0, \gamma \rangle$ where $\mathcal{S}$ is a finite set of states, $\mathcal{A}$ is a finite action set, $p : \mathcal{S} \times \mathcal{A} \to \Delta(\mathcal{S})$ is the transition probability function (where $\Delta(\cdot)$ denotes the probability simplex), $r : \mathcal{S} \times \mathcal{A} \to [0, R_{\max}]$ is the reward function, $d_0$ is the initial distribution of states, and $\gamma \in [0, 1)$ is the discount factor. The agent's goal is to learn a policy $\pi : \mathcal{S} \to \Delta(\mathcal{A})$ that maximizes the expected discounted sum of rewards. In this paper, we study the PG updates on expectation, not their stochastic variants. Thus, our presentation and analyses use the true gradient of the functions of interest. Below we formalize these concepts and we discuss representative algorithms in the trajectory and state-action formulations.

### 2.1 Trajectory Formulation

The expected discounted return can be defined as $J(\pi) = \mathbb{E}_{s_0, a_0, \ldots} \left[ \sum_{t=0}^{\infty} \gamma^t r(s_t, a_t) \right]$, where $s_0 \sim d_0, a_t \sim \pi(a_t|s_t)$, and $s_{t+1} \sim p(s_{t+1}|s_t, a_t)$. Moreover, $\tau$ denotes a specific trajectory, $\tau = \langle s_0, a_0, s_1, \ldots \rangle$, and $R(\tau)$ denotes the return of that trajectory, that is, $R(\tau) = \sum_{t=0}^{\infty} \gamma^t r(s_t, a_t)$.

PG methods seek the $\pi^*$ maximizing $J$, i.e., $\pi^* = \arg\max_\pi J(\pi) = \arg\max_\pi \int_\tau R(\tau)\pi(\tau) \; d\tau$. Policies often live in a restricted class $\Pi$ parameterized by $\theta \in \mathbb{R}^d$, and the problem becomes

$$\theta^* = \arg\max_\theta J(\pi_\theta) = \arg\max_\theta \int_\tau R(\tau)\pi_\theta(\tau) \; d\tau \; , \tag{1}$$

where $\pi_\theta(\tau)$ is the probability of $\tau$ under the policy indexed by $\theta$. Note that, although we write $\pi_\theta(\tau)$, policies define distributions over actions given states and they only indirectly define distributions over trajectories.

REINFORCE [27] is one of the most traditional PG methods. It computes, at each step, the gradient of $J(\pi_\theta)$ with respect to $\theta$ and performs the following update:

$$\theta_{t+1} = \theta_t + \epsilon_t \int_\tau \pi_{\theta_t}(\tau) R(\tau) \frac{\partial \log \pi_\theta(\tau)}{\partial \theta} \bigg|_{\theta=\theta_t} d\tau \; , \tag{2}$$

where $\epsilon_t$ is a stepsize. We shall replace $\pi_{\theta_t}$ by $\pi_t$ when the meaning is clear from context.

### 2.2 State-Action Formulation

In the state-action formulation, we use the standard notions of state and state-action value function:

$$V^\pi(s_t) = \mathbb{E}_{a_t, s_{t+1}, \ldots} \left[ \sum_{k=0}^{\infty} \gamma^k r(s_{t+k}, a_{t+k}) \right], \quad Q^\pi(s_t, a_t) = \mathbb{E}_{s_{t+1}, a_{t+1}, \ldots} \left[ \sum_{k=0}^{\infty} \gamma^k r(s_{t+k}, a_{t+k}) \right], \tag{3}$$

where $a_t \sim \pi(a_t|s_t)$, and $s_{t+1} \sim p(s_{t+1}|s_t, a_t)$, for $t \geq 0$. The policy gradient theorem [23] provides an update equivalent to the REINFORCE update in Equation 2 for the state-action formulation:

$$\theta_{t+1} = \theta_t + \epsilon \sum_s d^{\pi_t}(s) \sum_a \pi_t(a|s) Q^{\pi_t}(s, a) \frac{\partial \log \pi_\theta(a|s)}{\partial \theta} \bigg|_{\theta=\theta_t} \; , \tag{4}$$

where $d^\pi$ is the discounted stationary distribution induced by the policy $\pi$.

## 3 An Operator View of REINFORCE

The parameter updates in Eq. 2 and Eq. 4 involve the current policy $\pi_{\theta_t}$ in the sampling distribution (resp. the stationary distribution), the value function, and inside the log term. While the log term is generally easy to deal with, the non-stationarity of the sampling distribution and value function is the source of many difficulties in the optimization process. One possibility to alleviate this issue is to fix the sampling distribution and value function while only optimizing the parameters of the policy inside the log, an approach which finds its justification by devising a lower bound on $J$ [7, 8, 1] or a locally valid approximation [5, 18].

All these approaches can be cast as minimizing a divergence measure between the current policy $\pi$ and a fixed policy $\mu^1$ which achieves higher return than $\pi$. Thus, moving from $\pi$ to $\mu$ can be seen as a *policy improvement step* and we have $\mu = \mathcal{I}\pi$, with $\mathcal{I}$ the improvement operator. Since the resulting $\mu$ might not be in the set of realizable policies, the divergence minimization acts as a *projection step* using a projection operator $\mathcal{P}$. In all these cases, when only performing one gradient step to minimize the divergence, we recover the original updates of Eq. 2 and Eq. 4.

This decomposition in improvement and projection steps allows us to see PG methods not simply as performing gradient ascent on an objective function, but as the successive application of a *policy improvement* and a *projection operator*. It highlights, for example, that even if an improvement operator produces a $\mu$ with high returns, if the projection operator cannot approximate this $\mu$ well, then performance may not increase. That is, for a given projection operator, an improvement operator compatible with the projection may be preferred over an improvement operator that generates the most improved policies.

This makes us wonder which policies $\mu$ and which projections to use. In particular, we are interested in operators which satisfy the following two properties: (a) The optimal policy $\pi(\theta^*)$ should be a stationary point of the composition $\mathcal{P} \circ \mathcal{I}$, as iteratively applying $\mathcal{P} \circ \mathcal{I}$ would otherwise lead to a suboptimal policy, and (b) Doing an approximate projection step of $\mathcal{I}\pi$, using gradient ascent starting from $\pi$, should always lead to a better policy than $\pi$. In particular, if the combination leads to maximizing a function that is a lower bound of $J$ everywhere, we know the combination of the two steps, even when solved approximately, leads to an increase in $J$ and will converge to a locally optimal policy.

These tools allow us to explore several possibilities for these operators. In this section we present REINFORCE under the view of operators, in both trajectory and value-function formulations; and we discuss consequences and insights this perspective gives us. Later we discuss other PG methods under the operators perspective and how it sheds light on design choices made by these algorithms.

## 3.1 Trajectory Formulation

The proposition below formalizes, in the trajectory formulation, the idea of casting PG methods as the successive application of a policy improvement and a projection operator. It does so by presenting two operators that give rise to OP-REINFORCE, an operator version of REINFORCE [27].[2]

**Proposition 1.** *Assuming all returns $R(\tau)$ are positive, Eq. 2 can be seen as doing a gradient step to minimize $KL(R\pi_t||\pi)$ with respect to $\pi$, where $R\pi_t$ is the policy defined by*

$$R\pi_t(\tau) = \frac{1}{J(\pi_t)} R(\tau)\pi_t(\tau) . \tag{5}$$

*Hence, the two operators associated with OP-REINFORCE are:*

$$\mathcal{I}_\tau \pi(\tau) = R\pi(\tau) \quad , \quad \mathcal{P}_\tau \mu = \arg\min_{\pi \in \Pi} KL(\mu||\pi) , \tag{6}$$

*where $\Pi$ is the set of realizable policies.*

We prove this proposition and the following in the Appendix.

Even in the tabular case, $R\pi$ might not be achievable when the environment is stochastic and so the projection operator $\mathcal{P}_\tau$ is needed. Note that OP-REINFORCE is different from the original REINFORCE algorithm because it solves the projection exactly rather than doing just one step of gradient descent. Nevertheless, all stationary points of $J(\pi)$ are fixed points of OP-REINFORCE, which maintains the following important property:

**Proposition 2.** *$\pi(\theta^*)$ is a fixed point of $\mathcal{P}_\tau \circ \mathcal{I}_\tau$.*

## 3.2 State-Action Formulation

While a policy was defined as a distribution over trajectories in the trajectory formulation, it will be defined as a stationary distribution over states and action in the state-action formulation. Similar to the trajectory formulation, the policy improvement step can lead to policies which are not realizable.

We saw in Section 3.1 that doing the full projection implied by the operators leads to OP-REINFORCE, which is slightly different from REINFORCE . Similarly, although the policy gradient theorem states that the updates of Eq. 2 and Eq. 4 are identical, the resulting operators will be different:

**Proposition 3.** *If all $Q^\pi(s,a)$ are positive, Eq. 4 can be seen as doing a gradient step to minimize*

$$D_{V^{\pi_t}\pi_t}(Q^{\pi_t}\pi_t||\pi) = \sum_s d^{\pi_t}(s)V^{\pi_t}(s)KL(Q^{\pi_t}\pi_t||\pi) \ , \tag{7}$$

*where $D_{V^{\pi_t}\pi_t}$ and the distribution $Q^\pi\pi$ over actions are defined as*

$$D_z(\mu||\pi) = \sum_s z(s)KL\big(\mu(\cdot|s)||\pi(\cdot|s)\big) \ , \tag{8}$$

$$Q^\pi\pi(a|s) = \frac{1}{\sum_{a'} Q^\pi(s,a')\pi(a'|s)}Q^\pi(s,a)\pi(a|s) = \frac{1}{V^\pi(s)}Q^\pi(s,a)\pi(a|s) \ . \tag{9}$$

*Hence, the two operators associated with the state-action formulation are:*

$$\mathcal{I}_V\pi(s,a) = \left(\frac{1}{\mathbb{E}_\pi[V^\pi]}d^\pi(s)V^\pi(s)\right)Q^\pi\pi(a|s) \tag{10}$$

$$\mathcal{P}_V\mu = \arg\min_{z\in\Pi}\sum_s \mu(s)KL\big(\mu(\cdot|s)||z(\cdot|s)\big). \tag{11}$$

The improvement operator $\mathcal{I}_V$ affects both the distribution over states, where it increases the probabilities of states $s$ with large values $V(s)$, and the conditional distribution over actions given states, where it increases the probabilities of actions $a$ with large values $Q(s,a)$.

The projection operator is not the KL-divergence over the full distribution over state-action pairs. Rather, it treats each state independently, weighting them using the distribution over states of its first argument. In the tabular case, the optimum may be found immediately and is independent of the distribution over states, i.e. $\mathcal{P}_V\mu(a|s) = \mu(a|s)$.

**Proposition 4.** *$\pi(\theta^*)$ is a fixed point of $\mathcal{P}_V \circ \mathcal{I}_V$.*

Now that we derived operator versions of REINFORCE, further bridging the gap between policy-based and value-based methods, we study the properties of these operators.

## 3.3  $\mathcal{I}_\tau\pi$ can be arbitrarily close to $\pi$

In value-based methods, the Bellman optimality operator is well-known to be a $\gamma$-contraction where $\gamma$ is the discount factor, leading to a linear convergence rate of the value function in the tabular case [14, 3]. In contrast, for policy gradient methods, the improvement operator in the trajectory formulation, $\mathcal{I}_\tau$, can produce arbitrarily small improvements, as formalized in the proposition below.

**Proposition 5.** *The performance of the improved policy $\mathcal{I}_\tau\pi$ is given by*

$$J(\mathcal{I}_\tau\pi) = J(\pi)\left(1 + \frac{Var_\pi(R)}{(\mathbb{E}_\pi[R])^2}\right) \geq J(\pi). \tag{12}$$

If $\pi$ is almost deterministic and the environment is deterministic, then we have $\text{Var}_\pi(R) \approx 0$ and $J(\mathcal{I}_\tau\pi) \approx J(\pi)$. This result justifies the general intuition that deterministic policies can be dangerous for PG methods; not because they may perform poorly, but because they stall the learning process [17]. In that sense, entropy regularization can be seen as helping the algorithm make consistent progress. Moreover, note that the improvement operator $\mathcal{I}_\tau$ is weaker than the equivalent operator for value-based methods, which can be seen as a consequence of the smoothness of change in the policies of PG methods when compared to the abrupt changes that can occur in value-based methods.

## 3.4  A lower bound on the overall improvement

Although we established that the improvement operator leads to a policy achieving higher return, it could still be the case that the projection operator $\mathcal{P}$ annihilates all these gains, leading $\mathcal{P} \circ \mathcal{I}\pi$ to having a smaller expected return than $\pi$. The proposition below derives a lower bound for the difference in expected returns, proving this cannot be the case, even when the projection is not computed exactly.

**Proposition 6.** *For any two policies $\pi$ and $\mu$ such that the support of $\mu$ covers that of $\pi$, we have*

$$J(\pi) \geq J(\mu) + \mathbb{E}_\mu[V^\mu(s)][D_\mu(\mathcal{I}_V\mu||\mu) - D_\mu(\mathcal{I}_V\mu||\pi)] \tag{13}$$

$$= J(\mu) + \sum_s d^\mu(s) \sum_a Q^\mu(s,a)\mu(a|s) \log \frac{\pi(a|s)}{\mu(a|s)} \ . \tag{14}$$

*Hence, any policy $\pi$ such that $D_{\pi_t}(\mathcal{I}_V\pi_t||\pi) < D_{\pi_t}(\mathcal{I}_V\pi_t||\pi_t)$ implies $J(\pi) > J(\pi_t)$.*

While Eq. 13 makes explicit the relationship between the divergence with the improved policy and the expected return, Eq. 14 is reminiscent of the identity of Kakade and Langford [5]:

$$J(\pi) = J(\mu) + \sum_s d^\mu(s) \sum_a Q^\pi(s,a)[\pi(a|s) - \mu(a|s)] \ ,$$

but with $Q^\mu$ replacing $Q^\pi$, making it easier to estimate using samples from $\mu$, and $\log \pi(a|s)$ instead of $\pi(a|s)$, making the optimization easier. The price to pay, however, is the loss of the equality, replaced with a lower bound.

Fig. F.2 (Appendix F) compares our lower bound to the surrogate approximation used by conservative policy iteration (CPI) [5], which provides the theoretical motivation for TRPO [18] and PPO [20]. Although both are equivalent to first-order terms, matching the value and first derivative of $J$ for $\pi = \pi_t$, the CPI approximation is not a bound on the true objective and only guarantees improvement of the original objective for small stepsizes, unlike our global lower bound.

### 3.5  Optimal off-policy sampling distribution

When training an agent, it is commonly assumed that, when feasible, sampling trajectories from the current policy is better than using samples generated from another policy. We show here that this is not always the case and that, regardless of the current policy $\pi$, it can be beneficial to sample trajectories from another policy.

Applying $\mathcal{P} \circ \mathcal{I}$ is equivalent to minimizing the divergence between a fixed policy $\mathcal{I}\pi_t$ and the current policy $\pi$. The choice of $\mathcal{I}\pi_t$ stems from its guaranteed improvement over $\pi_t$ but, because $J(\mathcal{I}\mu) \geq J(\mu)$ for any policy $\mu$, we might try to find a good policy $\mu$ then minimize the divergence between $\mathcal{I}\mu$ and $\pi$. Looking at the formulation in Eq. 11, we see that this would be equivalent to repeatedly applying the policy gradient update, but to samples drawn from $\mu$ rather than the current policy. In that regard, the update would be equivalent to that of *off-policy* policy gradient methods, without correcting using importance weights, hence leading to a biased estimate of the gradient. Schaul et al. [17] partially explored this biased gradient, and report that off-policy sampling sometimes works better than on-policy sampling.

Because $\pi(\theta^*)$ is a fixed point of $\mathcal{P} \circ \mathcal{I}$, minimizing Eq. 11 with $\mu = \mathcal{I}\pi(\theta^*)$ will lead to the optimal policy. Hence, the optimal strategy, when the current policy is *any* policy $\pi$, is to draw samples from $\pi(\theta^*)$ instead and apply the biased policy gradient updates.[3] Although this result is of no practical interest since it requires knowing $\pi(\theta^*)$ in advance, it proves that there are better sampling distributions than the current policy and that, in some sense, off-policy learning without importance correction is "optimal" when sampling from the optimal policy.

## 4  Other policy gradient methods under the operators perspective

Now that we have shown that REINFORCE can be cast as iteratively applying two operators, we might wonder whether other operators could be used instead. In this section, we explore the use of other policy improvement and projection operators. We shall see that there are two main categories of transformation: the first one performs a nonlinear transformation of the rewards in the hope of reaching faster convergence; second changes the distribution over state-action pairs and possibly the ordering of the KL divergence. With this perspective, we recover operators that give rise to PPO [20] and MPO [1] to shed some light on what these methods truly accomplish.

### 4.1 Moving beyond returns

While REINFORCE improves the policy by weighting the policy by the returns, one might wonder if we can potentially substitute in other non-linear transformations of the return to speed up the learning process. Intuitively, policies at the beginning of training are usually of such poor quality that asking them to focus solely on the highest-return trajectories may lead to larger improvement steps. As such, one might wonder if policy improvement operators that transform the return can lead to faster convergence and, if so, whether the policy at convergence remains optimal. We discuss two such transformations that place increased emphasis on the highest-return trajectories.

#### 4.1.1 Polynomial returns

Since, in the trajectory formulation, the improvement step consists in multiplying the probability of each trajectory by its associated return, one might wonder what would happen if we instead used the return raised to the $k$-th power, i.e. replacing $\mathcal{I}_\tau$ by $\mathcal{I}_\tau^k : \pi \longrightarrow R^k \pi$. Larger values of $k$ place higher emphasis on high-return trajectories and, as $k$ grows to infinity, $R^k \pi$ becomes the deterministic distribution that assigns probability 1 to the trajectory achieving the highest return.[4] However, for any $k \neq 1$, $\pi(\theta^*)$ may not be a fixed point of $\mathcal{P}_\tau \circ \mathcal{I}_\tau^k$, since projecting a policy that achieves a higher expected return can still lead to a worse policy. Thankfully, the following proposition allows us to address the issue by changing the projection operator accordingly:

**Proposition 7.** *Let $\alpha \in (0, 1)$. Then $\pi(\theta^*)$ is a fixed point of $\mathcal{P}_\tau^\alpha \circ \mathcal{I}_\tau^{\frac{1}{\alpha}}$ with $\mathcal{P}_\tau^\alpha$ defined by*

$$\mathcal{P}_\tau^\alpha \mu = \arg \min_{\pi \in \Pi} D^\alpha(\mu || \pi) \, , \tag{15}$$

*where $D^\alpha$ is the $\alpha$-divergence or Rényi divergence of order $\alpha$.*

Proposition 7 is especially interesting in the context of imitation learning where the teacher distribution over trajectories is concentrated around few high performing trajectories. This distribution can be seen as $R^k \pi$ for an arbitrary $\pi$, say uniform, and a large value of $k$. We should then use an $\alpha$-divergence, not the KL, to recover a good policy. Ke et al. [6] pointed out the usefulness of $\alpha$-divergences in this context but we are not aware of previous connections with the fixed point property. Similarly, this is reminiscent of the bounds obtained by Ross and Bagnell [16] who show that combining the expert policy and the current policy, slowly giving more weight to the current policy, leads to tighter bounds.

We can use a similar approach for the state-action formulation, leading to the following proposition:

**Proposition 8.** *Let $\alpha \in (0, 1)$. Then $\pi(\theta^*)$ is a fixed point of $\mathcal{P}_V^\alpha \circ \mathcal{I}_V^{\frac{1}{\alpha}}$ with*

$$\mathcal{I}_V^\alpha \pi = (Q^\pi)^{\frac{1}{\alpha}} \pi \quad , \quad \mathcal{P}_{V,\pi}^\alpha \mu = \arg \min_{z \in \Pi} \sum_s d^\pi(s) Z_\mu^\pi(s) D^\alpha(\mu || z) \, , \tag{16}$$

*where $Z_\alpha^\pi(s) = \sum_a \pi(a|s) Q^\pi(s, a)^{\frac{1}{\alpha}}$ is a normalization constant.*

Note that, in the tabular case, because we are in the state-action formulation, we can ignore the projection operator and we get $\pi_t(a|s) \propto \pi_0(a|s) \left( \prod_{i=1}^{t-1} Q^{\pi_i}(s, a) \right)^{\frac{1}{\alpha}}$, and the policy becomes more deterministic as $\alpha$ goes to 0. In fact, at the limit $\alpha = 0$, $\mathcal{I}_V^\alpha \pi(\cdot|s)$ is the policy which assigns probability 1 to the action $a^*(s) = \arg \max_a Q^\pi(s, a)$, and $\mathcal{I}_V^\alpha$ becomes the greedy policy improvement operator. **The operator view can then be seen as offering us an interpolation between REINFORCE and some value-based methods such as Q-learning** [26]: we recover REINFORCE with $\alpha = 1$ and the Bellman optimality operator at the limit $\alpha = 0$. From this perspective, one may say that the main difference between REINFORCE and value-based methods is how aggressively they use value estimates to define their policy. REINFORCE generates smooth policies that choose actions proportionally to their estimated value, value-based methods choose the action with higher value.

**Empirical analysis**    Although using an $\alpha$-divergence is necessary to maintain $\pi(\theta^*)$ as stationary point, it is possible that using the KL will still lead to faster convergence early in training. We studied the effect of this family of improvement operators $\mathcal{I}^\alpha$ for different choices of $\alpha$ in the four-room domain [22] (Figure 1).The agent starts in the lower-left corner and seeks to reach the

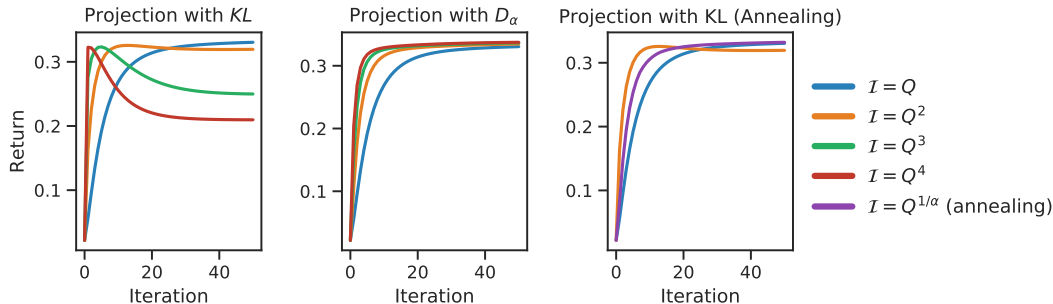

Figure 1: Evaluation of polynomial reward improvement operators $\mathcal{I}_V^{1/\alpha}$ paired with different projection steps in the four-room domain. The operator $\mathcal{I}_V^{1/\alpha}$ generally speeds up learning, but if paired with the KL projection (left), it can converge to a sub-optimal policy. If the improvement operator is paired with an $\alpha$-divergence (middle) or the value of $\alpha$ is annealed to 1 (right), learning is fast and converges to the optimal policy. Figure best seen in color.

upper-right corner; episode terminates with a reward of +1 upon entering the goal state. The policy is parameterized by a softmax and all states share the same parameters, i.e. we use function approximation.

When $\mathcal{I}_V^\alpha$ is paired with a KL projection step, the return increases faster than when using OP-REINFORCE's improvement operator early in the process. However, such a combination converges to a suboptimal policy and incurs linear regret (Figure 1). In contrast, combining the improvement operator with a projection that uses the corresponding $\alpha$-divergence not only speeds up learning but also converges to the optimal policy. One can use $\mathcal{I}^\alpha$ with the KL projection step heuristically, by selecting an aggressive improvement operator $\mathcal{I}^\alpha$ (low $\alpha$) early in optimization, and annealing $\alpha$ to 1 to recover OP-REINFORCE updates asymptotically. We present results of using line search to dynamically anneal the value of $\alpha$ as the policy converges (details in Appendix F.1).

### 4.1.2 Exponential returns

So far, all of our analysis assumed the returns were nonnegative. If the returns are lower bounded, this can be addressed by shifting them upward until they are all nonnegative. However, Le Roux [8] showed this is equivalent to adding a KL term that might slow down convergence. Another possibility is to transform these returns using a strictly increasing function with nonnegative outputs. The most common such transformation is the exponential function, leading to the operator $\mathcal{I}_\tau^{\exp,\beta}\pi = \exp\left(\beta R\right)\pi$, with $\beta > 0$. Although this transformation solves the non-negativity issue and makes the algorithm invariant to a shift in the rewards, similar to the original REINFORCE, we are not aware of any result guaranteeing that there is a projection operator such that $\pi(\theta^*)$ remains a fixed point. However, the following proposition shows that before applying a projection, the improved policy $\mathcal{I}_\tau^{\exp,\beta}\pi$ achieves a higher return than $\pi$. This is true, in fact, for any transformation using a strictly increasing function with nonnegative outputs.

**Proposition 9.** *Let $f$ be an increasing function such that $f(x) > 0$ for all $x$. Then*

$$J\big(f(R)\pi\big) = J(\pi) + \frac{Cov_\pi\big(R, f(R)\big)}{\mathbb{E}_\pi[f(R)]} \geq J(\pi). \tag{17}$$

While the improvement operator increases the return, we emphasize that any increase in performance may be annihilated by the consequent projection operator.

## 4.2 An operator view of PPO

PPO [20] is one of the most widely used policy gradient methods. At each iteration it maximizes a surrogate objective that depends on the current distribution over states and Q-function, which is clipped to avoid excessively large policy updates. The surrogate objective is to maximize $\sum_a \pi(a|s)Q^\mu(s,a)$, where states are sampled from the state distribution of $\mu$. PPO is often formulated with an entropy bonus and an entropy penalty coefficient $\beta > 0$ (letting $\beta \to \infty$ removes entropy regularization). The operators that allow us to recover PPO are presented below.

$$\mathcal{I}_V \pi(s,a) = d^\pi(s) \frac{\exp\left(\beta Q^\pi(s,a)\right)}{\sum_{a'} \exp\left(\beta Q^\pi(s,a')\right)} \tag{18}$$

$$\mathcal{P}_V \mu = \arg\min_{z \in \Pi} \sum_s \mu(s) KL\left(\text{clip}(z(\cdot|s)) || \mu(\cdot|s)\right) , \tag{19}$$

leading to 
$$\pi_{t+1} = \arg\min_z \sum_s d^{\pi_t}(s) \left( \sum_a z(a|s) Q^{\pi_t}(s,a) - \frac{1}{\beta} \sum_a z(a|s) \log z(a|s) \right) , \tag{20}$$

where we omitted the clipping on the last line for readability. There are three main differences between the operators that recover PPO and the operators of OP-REINFORCE (Eq. 10 and 11): (1) The policy improvement operator does not increase the probability of good states because, different from OP-REINFORCE, $V(s)$ is not part of $\mathcal{I}_V$; (2) the policy improvement operator only uses the $Q$-values in its distribution over actions given states, instead of also using $\pi$; (3) the KL in the projection operator is reversed. The last point is particularly important as this reversed KL is *mode seeking*, so the resulting distribution $\pi_{t+1}$ will focus its mass on the mode of $\mathcal{I}_V(\pi_t)$, which is the action with the largest $Q$-value. This can quickly lead to deterministic policies, especially when entropy regularization is not used, justifying the necessity of the clipping in the KL. By comparison, the projection operator of Eq. 11 uses a KL that is *covering*, naturally preventing $\pi_{t+1}$ from becoming too deterministic. While our analysis fits current PG methods into the operator view, they can also be framed in the language of optimization, for example as performing approximate mirror descent in an MDP [13].

### 4.3 An operator view of MPO

The operator view can also be used to provide insight on MPO [1], a state-of-the-art algorithm derived through the control-as-inference framework [4, 9]. The policy improvement operator that recovers MPO is:

$$\mathcal{I}_V \pi(s,a) = d^\pi(s) \frac{\pi(a,s) \exp\left(\beta Q^\pi(s,a)\right)}{\sum_{a'} \pi(a',s) \exp\left(\beta Q^\pi(s,a')\right)} , \tag{21}$$

with the projection operator being the same as OP-REINFORCE's. Note that the improvement operator interpolates between those of OP-REINFORCE and PPO: it does not upweight good states and it uses an exponential transformation of the rewards, like PPO, but it still uses the policy $\pi(a|s)$ and not just the rewards. In this case, clipping is not necessary because the KL is in the "covering" direction.

## 5 Related Work

Traditional analyses of PG methods cast the method as gradient ascent on the expected return objective. Using these tools, convergence can be shown to a stationary point [27, 18], and under sufficient assumptions about the function class, strengthened to convergence to the optimal policy [2]. When entropy regularization is also added, the optimal solutions of PG methods have been shown to coincide with those of soft Q-learning [11, 19]. This connection is generally limited to the optimal policy, and does not explain how the training dynamics of PG methods relate to value-based ones. In contrast, using operators enables a new perspective of the learning dynamics of PG methods, for example providing an interpolation between PG and value-based methods (Section 4.1.1).

Instead of following the REINFORCE update, many modern PG methods optimize a surrogate approximation, the most common being the linear surrogate expansion of the expected return by Kakade and Langford [5], which can be generalized to a higher-order Taylor expansion [24]. While the lower bound surrogate objective derived with the operator view (Proposition 6) agrees with these approximations to first-order terms, they have different behavior globally. Most prior surrogate objectives penalize deviation from the original data-collection policy, which causes PG methods to take excessively conservative steps, whereas the operator lower bound penalizes deviation from the *improved* policy, potentially avoiding this issue.

The operator view is closely related to control-as-inference, a line of work that casts RL as performing inference in a graphical model [4, 9], where the return of a trajectory corresponds to an unnormalized

log probability. EM algorithms in this graphical model are analogous to OP-REINFORCE, where the expectation step corresponds to policy improvement, and the maximization step corresponds to projection. Similarly, incomplete projection steps by partially maximizing the lower bound in Proposition 6 is equivalent to doing incremental EM [12]. While the operator view has a clear semantic interpretation, the semantics of the graphical model in control-as-inference are controversial in many contexts, since rewards in most MDPs cannot easily be interpreted as probabilities [9]. The operator view is also more general, since it allows us to incorporate RL algorithms like PPO that do not fit cleanly into the control-as-inference formulation.

## 6    Conclusion

We cast PG methods as the repeated application of two operators: a policy improvement operator and a projection operator. Starting with a modification of REINFORCE, we introduced the operators that recover well-known algorithms such as PPO and MPO. This operator perspective also allowed us to further bridge the gap between policy-based and value-based methods, showing how REINFORCE and the Bellman optimality operator can be seen as the same method with only one parameter changing.

Importantly, this perspective helps us improve our understanding behind decisions often made in the field. We showed how entropy regularization helps by increasing the variance of the returns, guaranteeing larger improvements for policy methods; we showed how even single gradient steps towards the full projection operator are guaranteed to lead to an improvement; and how practices such as exponentiating rewards to make them non-negative, as done by MPO, still lead to a meaningful policy improvement operator. Finally, by introducing new operators based on the $\alpha$-divergence we were able to show that there are other operators that can still lead to faster learning, shedding some light into how to better use, for example, expert trajectories in reinforcement learning algorithms, as often done in high-profile success stories [21, 25].

Finally, we hope the results we presented in this paper will empower researchers to design new policy gradient methods, either through the introduction of new operators, or by leveraging the intuitions we presented here. This operator perspective opens up a new avenue of research in analyzing policy gradient methods and it can also provide a different perspective on traditional problems in the field, such as how to choose appropriate basis functions to better represent policies, and how to do better exploration by design sampling policies different than the agent's current policy.

### Acknowledgements

We thank Doina Precup, Dale Schuurmans, Philip Thomas, Marc G. Bellemare, Kavosh Asadi, Danny Tarlow, and members of the Brain Montreal team for enlightening discussions and feedback on earlier drafts of the paper. NLR is supported by a Canada CIFAR AI Chair.

### Broader Impact

As this work has a theoretical focus, it is unlikely to have a direct impact on society at large although it may guide future research with such an impact.

## Footnotes

[1]"Policy" is loosely defined here as a distribution over trajectories.

[2]OP-REINFORCE was originally introduced by Le Roux [8] under the name "Iterative PoWER".

[3]Assuming we can find the optimum of the projection, which is the case when the class of policies belongs to the exponential family.

[4]If there are multiple such trajectories, this is a uniform distribution over all of them.

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
