[Supplementary Material]

In the entirety of the Appendix, we shall use $\pi^*$ instead of $\pi(\theta^*)$ for increased readability.

## A  Definition of the operators

**Proposition 1.** *Assuming all returns $R(\tau)$ are positive, Eq. 2 can be seen as doing a gradient step to minimize $KL(R\pi_t||\pi)$ with respect to $\pi$, where $R\pi_t$ is the policy defined by*

$$R\pi_t(\tau) = \frac{1}{J(\pi_t)} R(\tau)\pi_t(\tau) . \tag{5}$$

*Hence, the two operators associated with* OP-REINFORCE *are:*

$$\mathcal{I}_\tau \pi(\tau) = R\pi(\tau) \quad , \quad \mathcal{P}_\tau \mu = \arg\min_{\pi \in \Pi} KL(\mu||\pi) , \tag{6}$$

*where $\Pi$ is the set of realizable policies.*

*Proof.* Denoting $\mu$ the distribution over trajectories such that $\mu(\tau) \propto R(\tau)\pi(\tau)$, we have

$$KL(\mu||\pi) = \int_\tau \mu(\tau) \log \frac{\mu(\tau)}{\pi(\tau)} \, d\tau \tag{22}$$

$$\frac{\partial KL(\mu||\pi)}{\partial \theta} = -\int_\tau \mu(\tau)\nabla_\theta \log \pi(\tau) \, d\tau \tag{23}$$

$$\propto -\int_\tau R(\tau)\pi(\tau)\nabla_\theta \log \pi(\tau) \, d\tau \tag{24}$$

by definition of $\mu(\tau)$. $\qquad\square$

**Proposition 3.** *If all $Q^\pi(s,a)$ are positive, Eq. 4 can be seen as doing a gradient step to minimize*

$$D_{V^{\pi_t}\pi_t}(Q^{\pi_t}\pi_t||\pi) = \sum_s d^{\pi_t}(s)V^{\pi_t}(s)KL(Q^{\pi_t}\pi_t||\pi) , \tag{7}$$

*where $D_{V^{\pi_t}\pi_t}$ and the distribution $Q^\pi\pi$ over actions are defined as*

$$D_z(\mu||\pi) = \sum_s z(s)KL\big(\mu(\cdot|s)||\pi(\cdot|s)\big) , \tag{8}$$

$$Q^\pi\pi(a|s) = \frac{1}{\sum_{a'} Q^\pi(s,a')\pi(a'|s)} Q^\pi(s,a)\pi(a|s) = \frac{1}{V^\pi(s)} Q^\pi(s,a)\pi(a|s) . \tag{9}$$

*Hence, the two operators associated with the state-action formulation are:*

$$\mathcal{I}_V \pi(s,a) = \left( \frac{1}{\mathbb{E}_\pi[V^\pi]} d^\pi(s)V^\pi(s) \right) Q^\pi\pi(a|s) \tag{10}$$

$$\mathcal{P}_V \mu = \arg\min_{z \in \Pi} \sum_s \mu(s)KL\big(\mu(\cdot|s)||z(\cdot|s)\big). \tag{11}$$

*Proof.*

$$\sum_s d^{\pi_t}(s)V^\pi(s)\frac{\partial KL(Q^\pi\pi_t||\pi)}{\partial \theta} = -\sum_s d^{\pi_t}(s)V^\pi(s)\sum_a Q^\pi\pi_t(a|s)\nabla_\theta \log \pi(a|s)$$

$$= -\sum_s d^{\pi_t}(s)\sum_a Q^\pi(s,a)\pi_t(a|s)\nabla_\theta \log \pi(a|s) , \tag{25}$$

and we recover the update of Eq. 4. $\qquad\square$

# B  $\pi^*$ is a stationary point when using the KL

**Proposition 2.** $\pi(\theta^*)$ *is a fixed point of* $\mathcal{P}_\tau \circ \mathcal{I}_\tau$.

*Proof.* We have

$$\nabla_\theta KL(R\pi^* \| \pi)\Big|_{\pi=\pi^*} = \int_\tau R(\tau)\pi^*(\tau)\nabla_\theta \log \pi^*(\tau)\, d\tau \tag{26}$$

$$= 0 \text{ by definition of } \pi^* . \tag{27}$$

$\square$

**Proposition 4.** $\pi(\theta^*)$ *is a fixed point of* $\mathcal{P}_V \circ \mathcal{I}_V$.

*Proof.* We have

$$\nabla_\theta \sum_s d^{\pi^*}(s)V^{\pi^*}(s)KL(Q^{\pi^*}\pi^* \| \pi)\Big|_{\pi=\pi^*} = \sum_s d^{\pi^*}(s)\sum_a \pi^*(a|s)Q^{\pi^*}(s,a)\frac{\partial \log \pi_\theta(a|s)}{\partial\theta}\Big|_{\theta=\theta^*} \tag{28}$$

$$= 0 \text{ by definition of } \pi^* . \tag{29}$$

$\square$

# C  Expected return of the improved policy

We use the same proof for the following two propositions:

**Proposition 5.** *The performance of the improved policy* $\mathcal{I}_\tau\pi$ *is given by*

$$J(\mathcal{I}_\tau\pi) = J(\pi)\left(1 + \frac{Var_\pi(R)}{(\mathbb{E}_\pi[R])^2}\right) \geq J(\pi). \tag{12}$$

**Proposition 9.** *Let* $f$ *be an increasing function such that* $f(x) > 0$ *for all* $x$. *Then*

$$J\big(f(R)\pi\big) = J(\pi) + \frac{Cov_\pi\big(R, f(R)\big)}{\mathbb{E}_\pi[f(R)]} \geq J(\pi). \tag{17}$$

*Proof.* We now show the expected return of the policy $z\pi$, defined as

$$z\pi(\tau) = \frac{1}{\int_{\tau'} z(\tau')\pi(\tau')\, d\tau'} z(\tau)\pi(\tau) , \tag{30}$$

for any function $z$ over trajectories. In particular, we show that choosing $z = R$ leads to an improvement in the expected return.

$$J(z\pi) = \int_\tau R(\tau)(z\pi)(\tau)\, d\tau \tag{31}$$

$$= \int_\tau \frac{R(\tau)z(\tau)\pi(\tau)}{\int_{\tau'} z(\tau')\pi(\tau')\, d\tau'}\, d\tau \tag{32}$$

$$= \left(\int_{\tau'} R(\tau')\pi(\tau')\, d\tau'\right)\frac{\int_\tau R(\tau)z(\tau)\pi(\tau)\, d\tau}{\int_{\tau'} z(\tau')\pi(\tau')\, d\tau' \int_{\tau'} R(\tau')\pi(\tau')\, d\tau'} \tag{33}$$

$$= J(\pi)\frac{\mathbb{E}_\pi[Rz]}{\mathbb{E}_\pi[R]\mathbb{E}_\pi[z]} \tag{34}$$

$$= J(\pi)\left(1 + \frac{Cov_\pi(R, z)}{\mathbb{E}_\pi[R]\mathbb{E}_\pi[z]}\right) , \tag{35}$$

where $Cov_\pi(R, z) = \mathbb{E}_\pi[Rz] - \mathbb{E}_\pi[R]\mathbb{E}_\pi[z]$.

When $z = R$, the expected return becomes

$$J(R\pi) = J(\pi) \left( 1 + \frac{\text{Var}_\pi(R)}{(\mathbb{E}_\pi[R])^2} \right) \tag{36}$$

$$\geq J(\pi) . \tag{37}$$

$\square$

# D $\quad \pi^*$ is a stationary point when using $\alpha$-divergence

**Proposition 7.** *Let $\alpha \in (0, 1)$. Then $\pi(\theta^*)$ is a fixed point of $\mathcal{P}_\tau^\alpha \circ \mathcal{I}_\tau^{\frac{1}{\alpha}}$ with $\mathcal{P}_\tau^\alpha$ defined by*

$$\mathcal{P}_\tau^\alpha \mu = \arg \min_{\pi \in \Pi} D^\alpha(\mu || \pi) , \tag{15}$$

*where $D^\alpha$ is the $\alpha$-divergence or Rényi divergence of order $\alpha$.*

*Proof.* We now show that $\pi^*$ is the fixed point of $\mathcal{P}_\tau^\alpha \circ \mathcal{I}_\tau^\alpha$. The minimizer of $d^\alpha$ with respect to its second argument can be computed through iterative minimization of $D_{\alpha'}$ for any other nonzero $\alpha'$ [10]:

$$z_{t+1} = \arg \min_z D_{\alpha'} \left( \pi^{\alpha/\alpha'} z_t^{1 - \alpha/\alpha'} \big|\big| z \right) . \tag{38}$$

In the remainder of this proof, we shall use $\alpha' = 1$, leading to

$$z_{t+1} = \arg \min_z KL(\pi^\alpha z_t^{1-\alpha} || z). \tag{39}$$

We know that $\pi^*$ is a stationary point of $\mathcal{P}_\tau^1 \circ \mathcal{I}_\tau^1$, i.e.

$$\pi^* = \arg \min_z KL(R\pi^* || z). \tag{40}$$

Hence, we see that, if $\pi^\alpha(\pi^*)^{1-\alpha} = R\pi^*$, the iterative process described in Eq. 39 initialized with $z_0 = \pi^*$ will be stationary with $z_i = \pi^*$ for all $i$. This gives us the form we need for $\pi = \mathcal{I}_\tau^\alpha \pi^*$. Indeed, we must have

$$\pi^\alpha(\pi^*)^{1-\alpha} = R\pi^* \tag{41}$$

$$(\mathcal{I}^\alpha \pi^*)^\alpha (\pi^*)^{1-\alpha} = R\pi^* \tag{42}$$

$$\mathcal{I}^\alpha \pi^* = [R(\pi^*)^\alpha]^{1/\alpha} \tag{43}$$

$$= R^{1/\alpha} \pi^* \tag{44}$$

$$\mathcal{I}^\alpha = (\pi \longrightarrow R^{1/\alpha}\pi). \tag{45}$$

$\square$

**Proposition 8.** *Let $\alpha \in (0, 1)$. Then $\pi(\theta^*)$ is a fixed point of $\mathcal{P}_V^\alpha \circ \mathcal{I}_V^{\frac{1}{\alpha}}$ with*

$$\mathcal{I}_V^\alpha \pi = (Q^\pi)^{\frac{1}{\alpha}} \pi \quad , \quad \mathcal{P}_{V,\pi}^\alpha \mu = \arg \min_{z \in \Pi} \sum_s d^\pi(s) Z_\mu^\pi(s) D^\alpha(\mu || z) , \tag{16}$$

*where $Z_\alpha^\pi(s) = \sum_a \pi(a|s) Q^\pi(s, a)^{\frac{1}{\alpha}}$ is a normalization constant.*

*Proof.* The proof is very similar to that of Proposition 7. We know that $\pi^*$ is a stationary point of $\mathcal{P}_V^1 \circ \mathcal{I}_V^1$, i.e.

$$0 = \sum_s d^{\pi^*}(s) \sum_a \pi^*(a|s) Q^{\pi^*}(s, a) \frac{\partial \log \pi_\theta(a|s)}{\partial \theta} \bigg|_{\theta = \theta^*} \tag{46}$$

$$= \sum_s d^{\pi^*}(s) \sum_a \pi^*(a|s)^{1-\alpha} \left( \pi^*(a|s) Q^{\pi^*}(s, a)^{\frac{1}{\alpha}} \right)^\alpha \frac{\partial \log \pi_\theta(a|s)}{\partial \theta} \bigg|_{\theta = \theta^*} \tag{47}$$

$$= \sum_s d^{\pi^*}(s) Z_\alpha(s) \nabla_\theta KL \left( \left( \pi^*(\cdot|s) Q^{\pi^*}(s, \cdot)^{\frac{1}{\alpha}} \right)^\alpha \pi^*(a|s)^{1-\alpha} \big|\big| \pi \right) \bigg|_{\pi = \pi^*} , \tag{48}$$

where $Z_\alpha(s) = \sum_a \pi^*(\cdot|s)Q^{\pi^*}(s,\cdot)^{\frac{1}{\alpha}}$ is the normalization constant. Hence, each iteration of Eq. 39 will leave $\pi^*$ unchanged.

Hence, we see that, if $\pi^\alpha(\pi^*)^{1-\alpha} = R\pi^*$, the iterative process described in Eq. 39 initialized with $z_0 = \pi^*$ will be stationary with $z_i = \pi^*$ for all $i$. This gives us the form we need for $\pi = \mathcal{I}^\alpha \pi^*$. Indeed, we must have

$$\pi^\alpha(\pi^*)^{1-\alpha} = Q\pi^* \tag{49}$$

$$(\mathcal{I}^\alpha \pi^*)^\alpha (\pi^*)^{1-\alpha} = Q\pi^* \tag{50}$$

$$\mathcal{I}^\alpha \pi^* = [Q(\pi^*)^\alpha]^{1/\alpha} \tag{51}$$

$$= Q^{1/\alpha}\pi^* \tag{52}$$

$$\mathcal{I}^\alpha = (\pi \longrightarrow Q^{1/\alpha}\pi). \tag{53}$$

$\square$

# E   Lower bounds

## E.1   Trajectory formulation

We state here the proposition for the trajectory formulation.

**Proposition 10** (Trajectory formulation)**.** *For any two distributions $\pi$ and $\mu$, we have*

$$J(\pi) \geq J(\mu)\left(1 - KL(\mathcal{I}_\tau \mu||\pi) + KL(\mathcal{I}_\tau \mu||\mu)\right). \tag{54}$$

*Hence, any policy $\pi$ such that $KL(\mathcal{I}_\tau \pi_t||\pi) < KL(\mathcal{I}_\tau \pi_t||\pi_t)$ implies $J(\pi) > J(\pi_t)$.*

*Proof.* Let $\pi$ and $\mu$ be two arbitrary distributions over trajectories such that the support of $\pi$ is included in that of $\mu$. Then

$$J(\pi) = \int_\tau R(\tau)\pi(\tau)\,d\tau \tag{55}$$

$$= \int_\tau R(\tau)\frac{\pi(\tau)}{\mu(\tau)}\mu(\tau)\,d\tau \tag{56}$$

$$\geq \int_\tau R(\tau)\left(1 + \log\frac{\pi(\tau)}{\mu(\tau)}\right)\mu(\tau)\,d\tau \tag{57}$$

$$= \int_\tau R(\tau)\mu(\tau)\,d\tau + \int_\tau R(\tau)\mu(\tau)\log\pi(\tau)\,d\tau - \int_\tau R(\tau)\mu(\tau)\log\mu(\tau)\,d\tau \tag{58}$$

$$= J(\mu) - J(\mu)KL(R\mu||\pi) + J(\mu)KL(R\mu||\mu) \tag{59}$$

$$J(\pi) \geq J(\mu)\left(1 - KL(R\mu||\pi) + KL(R\mu||\mu)\right). \tag{60}$$

$\square$

## E.2   State-action formulation

To prove that minimizing Eq. 7 is equivalent to maximizing a lower bound on the expected return $J$, we shall show that this function has the same gradient as a lower bound $J_\mu$ on $J$ and thus only differs by a constant.

**Proposition 11.** *Let us define $J_\mu$ as*

$$J_\mu(\pi) = \sum_{h=0}^{H} \gamma^h \int_\tau r(s_h, a_h)\left(1 + \log\frac{\pi_h(\tau_h)}{\mu_h(\tau_h)}\right)\mu(\tau)\,d\tau, \tag{61}$$

*where $H$ is the horizon (which can be infinite), $\tau_h$ is the trajectory of length $h$ that is a prefix of the full trajectory $\tau$, and $\pi_h(\tau_h)$ (resp. $\mu_h(\tau_h)$) is the total probability mass of trajectories with prefix $\tau_h$ under policy $\pi$ (resp. $\mu$).*

*Then we have $J_\mu(\pi) \leq J(\pi)$ for any $\mu$ and any $\pi$ such that the support of $\mu$ covers that of $\pi$.*

*Proof.* We can rewrite

$$J(\pi) = \int_\tau R(\tau)\pi(\tau)\,d\tau \tag{62}$$

$$= \int_\tau \left(\sum_{h=0}^{H} \gamma^h r(s_h, a_h)\right)\pi(\tau)\,d\tau \tag{63}$$

$$= \sum_{h=0}^{H} \gamma^h \int_\tau r(s_h, a_h)\pi(\tau)\,d\tau \tag{64}$$

$$= \sum_{h=0}^{H} \gamma^h \int_\tau r(s_h, a_h)\pi_h(\tau_h)\,d\tau. \tag{65}$$

Then, using the same technique as for the trajectory formulation, we have

$$J(\pi) = \sum_{h=0}^{H} \gamma^h \int_\tau r(s_h, a_h)\frac{\pi_h(\tau_h)}{\mu_h(\tau_h)}\mu_h(\tau_h)\,d\tau \tag{66}$$

$$\geq \sum_{h=0}^{H} \gamma^h \int_\tau r(s_h, a_h)\left(1 + \log\frac{\pi_h(\tau_h)}{\mu_h(\tau_h)}\right)\mu_h(\tau_h)\,d\tau \tag{67}$$

$$= \sum_{h=0}^{H} \gamma^h \int_\tau r(s_h, a_h)\left(1 + \log\frac{\pi_h(\tau_h)}{\mu_h(\tau_h)}\right)\mu(\tau)\,d\tau \tag{68}$$

$$= J_\mu(\pi). \tag{69}$$

□

Then we can prove the following proposition:

**Proposition 6.** *For any two policies $\pi$ and $\mu$ such that the support of $\mu$ covers that of $\pi$, we have*

$$J(\pi) \geq J(\mu) + \mathbb{E}_\mu[V^\mu(s)][D_\mu(\mathcal{I}_V\mu||\mu) - D_\mu(\mathcal{I}_V\mu||\pi)] \tag{13}$$

$$= J(\mu) + \sum_s d^\mu(s) \sum_a Q^\mu(s, a)\mu(a|s)\log\frac{\pi(a|s)}{\mu(a|s)}. \tag{14}$$

*Hence, any policy $\pi$ such that $D_{\pi_t}(\mathcal{I}_V\pi_t||\pi) < D_{\pi_t}(\mathcal{I}_V\pi_t||\pi_t)$ implies $J(\pi) > J(\pi_t)$.*

*Proof.* Since $J_\mu$ is a lower bound on $J$, by Proposition 11, we prove that its gradient is the same as that of

$$\nabla_\theta J_\mu(\pi) = \nabla_\theta\left(\sum_{h=0}^{H} \gamma^h \int_\tau r(s_h, a_h)\left(1 + \log\frac{\pi_h(\tau_h)}{\mu_h(\tau_h)}\right)\mu(\tau)\,d\tau\right) \tag{70}$$

$$= \sum_{h=0}^{H} \gamma^h \int_\tau r(s_h, a_h)\nabla_\theta\log\pi_h(\tau_h)\mu(\tau)\,d\tau \tag{71}$$

$$= \sum_{h=0}^{H} \gamma^h \int_\tau r(s_h, a_h)\left(\sum_{h'=0}^{h} \nabla_\theta\log\pi(a_{h'}|s_{h'})\right)\mu(\tau)\,d\tau \tag{72}$$

$$= \sum_{h'=0}^{H} \int_\tau \nabla_\theta\log\pi(a_{h'}|s_{h'})\left(\sum_{h=h'}^{H} \gamma^h r(s_h, a_h)\right)\mu(\tau)\,d\tau \tag{73}$$

$$= \sum_{h'=0}^{H} \sum_s \sum_a \nabla_\theta\log\pi(a|s)d_\mu^{h'}(s)\mu(a|s)\gamma^{h'}Q^\mu(s, a) \tag{74}$$

$$= \sum_{h'=0}^{H} \gamma^{h'} \sum_s d_\mu^{h'}(s) \sum_a \nabla_\theta\log\pi(a|s)\mu(a|s)\gamma^{h'}Q^\mu(s, a) \tag{75}$$

$$= \sum_s d^\mu(s) \sum_a Q^\mu(s,a)\mu(a|s)\nabla_\theta \log \pi(a|s) \tag{76}$$

$$= \nabla_\theta \left( -\sum_s d^\mu(s)V^\mu(s)KL(Q^\mu\mu||\pi) \right). \tag{77}$$

Hence these two functions only differ by a constant. Using $J_\pi(\pi) = \pi$, we identify the constant as being $J(\mu) + \mathbb{E}_\mu[V^\mu(s)]D_{\mathcal{I}_\mu}(\mu)$. $\qquad\square$

# F   Experimental Details

We reiterate the details of our didactic empirical study in the four-room domain [22]. An agent starts in the lower-left corner and seeks to reach the upper-right corner; upon entering the goal state, the agent receives a reward of +1 and terminates the episode. The policy is parameterized by softmax probabilities, $\pi_\theta(a|s) = \frac{\exp(\theta_a)}{\sum_{a\in\mathcal{A}}\exp(\theta_a)}$ for $\theta \in \mathbb{R}^{|\mathcal{A}|}$, where all states share the same parameters. As with our analysis, these experiments compute gradients and operators exactly; in practice, stochasticity from sampling and approximate value estimation can affect the resultant performance. In Figure F.2, we plot policies in the sub-segment $\{[0.1, 0.8t, 0.8(1-t), 0.1] : t \in [0,1]\}$, denoting the probability of taking the down, left, up, and right actions respectively.

The linear approximation of conservative policy iteration (CPI) [5] presented in Fig. F.2 does not include the quadratic term in $\alpha$ necessary to maintain the lower bound property (see Theorem 1 of [18]) as this term can be made arbitrarily large by setting $\gamma$ arbitrarily close to 1. This makes algorithms like TRPO very conservative when optimizing the lower bound, leading them to optimize relaxations instead.

Figure F.2: We visualize the true objective $J(\pi)$, the operator lower bound (Proposition 6), and the linear approximation optimized by TRPO and PPO on a 1d subspace of the policy space for the four-room domain (details in Appendix F). On the left, we plot the auxiliary objectives corresponding to different choices of sampling $\pi_t$. On the right, the objectives are plotted locally around $\pi_t$.

## F.1   Multi-step operators with line search

Proposition 6 implies that the single-step improvement operator converges to a desired solution by demonstrating that it fully minimizes a lower bound on the expected return. As partial minimization of this lower-bound also implies convergence, we propose a line-search approach that chooses the minimum $\alpha$ under which the lower-bound is optimized. Specifically, letting $L_\mu(\pi)$ be the lower bound in Proposition 6, we choose the lowest alpha such that

$$L_\mu(\mu) - L_\mu(\mathcal{PI}^\alpha\mu) \geq \frac{1}{2}\left(L_\mu(\mu) - L_\mu(\mathcal{PI}^1\mu)\right) \tag{78}$$