[Reviews · NeurIPS 2020]

Review 1

Summary and Contributions: The paper proposes to model the policy gradient updates as the sequential application of two operators: an improvement operator and a projection operator. The improvement operator, given the current policy, provides policy with non-decreased performance. The projection operator recovers a parametric representation of the obtained policy in the considered policy space, still preserving the non-decreased performance. Several improvements and projection operators are presented and put in connection with existing algorithms. Some of their properties are studied, like the fact that the optimal policy is a fixed point and a monotonic improvement result.

Strengths: I think that the paper provides a very interesting view of policy search by means of operators. The main point of strength of the paper is novelty. To the best of my knowledge, this is the first paper that provides these kind of insights and I think it might open new appealing research directions. I particularly liked the fact that the authors introduce a class of operators that interpolate between policy-based and value-based methods (Section 4.1.1).

Weaknesses: - In Section 2, it is assumed that the state and action spaces are finite. Is this assumption really necessary? It might be quite limiting since policy gradient methods are typically employed when dealing with continuous state-action spaces. Moreover, the transition model is defined as deterministic. Is this assumption necessary? - Proposition 6: I am a little confused about the notation. Equation (13) employs the improvement operator for the value-based case, but the remark in the subsequent line is stated for the improvement operator for the trajectory-based case. - Proposition 5: This is more curiosity than an issue. Are there some sufficient conditions to enforce that Var(R) > 0 along the whole learning process? It seems to me that as we get close to the optimum we are going to prefer less stochastic policies, thus we slow down convergence. Do you think we can still converge asymptotically if deterministic policies are allowed? Anyway, in policy search, we could even limit to stochastic policies. Maybe in such a case, we can have a non-zero guaranteed improvement and, consequently, converge in a finite number of iterations. I think the paper would greatly benefit from a discussion on these points. - Proposition 2: This is also a curiosity. The optimal policy, in the considered policy space, is a fixed point of the operator. There can be other fixed points? If so, do the authors think that is possible to characterize the space of fixed-points? Are there some conditions under which the fixed point is unique? ***Minor*** - lines 50 and 62: s_{t_1} -> s_{t+1} - Equation (3) goes beyond margins - line 64: reporting the formal definition of d^\pi might help - Equation (24) there should be a \propto instead of = - Proposition 5: there should be a statement, not just a formula - The notation of Proposition 9 does not match that used in the proof (z vs f(R)) - Figure 2: not very readable in grayscale, I suggest using different linestyles or markers

Correctness: I made a high-level check of the proofs in the appendix and everything seems correct to me.

Clarity: The paper is written in good English and reads well.

Relation to Prior Work: The paper explores a direction that is quite novel. Nevertheless, the authors make adequate connections with existing works, rephrasing them in their operator-based framework.

Reproducibility: Yes

Additional Feedback: I think that provides a valuable contribution to NeurIPS. I am also convinced that the paper can be strengthened if more insights on the theoretical properties of the operators were discussed (see for instance my last two points in the "Weaknesses" section). Nevertheless, I realize that the paper addressed an unexplored view of policy gradients and I am happy to reward its novelty. ***Post Rebuttal*** I thank the authors for their feedback. I have read it together with the other reviews. I think that my concerns have been effectively addressed. I agree with the other reviewers that the clarity can be improved and I encourage the authors to do so, but I am confirming my initial evaluation.


Review 2

Summary and Contributions: In this paper, the authors provide an operator view of the family of policy gradient methods in reinforcement learning. They show that in general, each update step in the policy gradient methods can be broken up into the application of two operators, a policy improvement operator, and a projection operation. This view bridges the gap between policy- and value-based learning algorithms. A new global lower bound is provided in the paper.

Strengths: The operator view of policy gradient methods proposed in this paper is significant and novel. It provides a unified framework and new perspective for understanding existing reinforcement learning methods as well as theoretical guidance for designing new algorithms. Thus it is highly relevant to researchers in the reinforcement learning community. It is also very interesting that the framework can lead to a unified view of policy- and value-based learning algorithms and induce interpolated algorithms. Besides, the paper is well-organized. The arguments, explanations, and examples are arranged in a coherent fashion. Reading the paper is really an enjoyable experience.

Weaknesses: - Lack of a dedicated related work section. - Lack of more realistic experiments.

Correctness: yes

Clarity: yes

Relation to Prior Work: yes

Reproducibility: Yes

Additional Feedback: - Although most of the related works are discussed in the intro section, it would help readers understand the position of this paper better if there is a dedicated related work section. - It would be interesting to more empirically analyze the induced interpolated algorithm between policy- and value-based methods under Proposition 8, by comparing algorithms induced by different \alphas in [0,1] on more realistic benchmark tasks, followed by a detailed discussion on their behaviors. - Alphago-zero[1] also fits into the framework where they use Monte Carlo Tree Search as the policy improvement operator. It's worth mentioning that in the discussion as well. - Typo in line 43, page 2. "learn a policy \pi : S x A --> \delta(S)" should be "learn a policy \pi : S --> \delta(A)". [1] Silver, D., Schrittwieser, J., Simonyan, K., Antonoglou, I., Huang, A., Guez, A., ... & Chen, Y. (2017). Mastering the game of go without human knowledge. nature, 550(7676), 354-359. Edit: As mentioned by R3, the writing of the paper lacks clarity and not being addressed in the response. Therefore my score is updated to 6 and I expect a major improvement on the writing of the paper.


Review 3

Summary and Contributions: The paper reinterprets policy improvement as applying an improvement operator and a projection operator. This approach offers new theoretical insights into existing policy gradient algorithms as well as the connection between value and policy based methods.

Strengths: The paper presents an interesting perspective on classic policy gradients as maximising a KL divergence. Although this has been observed before in the RL as inference framework, the formulation here is more general. The authors claim this affords several new insights into existing policy gradient methods, but I'm struggling to understand what these are.

Weaknesses: The paper is full of errors, so much so that I don't have enough space in the 'correctness' part of my review to write them all down (I only got to the end of the second page before the text box was full). It is very difficult to gain any form of motivation for the proposed perspective due to the major issues in presentation and errors in the text. To me, it seems like what is being proposed is very similar to the pseudo-likelihood approach of RL as inference, albeit with a slightly different objective that offers a form of generalisation. Whilst this may be interesting, what is presented here feels like a series of mathematical results and I cannot understand how it relates to a bigger picture. Many of the paragraphs of discussion where key insights are given are incomprehensible to me.

Correctness: Line 19: 'In particular, value-based methods are often cast as the iterative application of a policy improvement operator, the Bellman optimality operator, which transforms the value function into a “better” one (unless the value function is already the optimal one)' Value-based RL methods such as Q learning and SARSA don't involve policy improvement. They apply policy evaluation. SARSA does not apply the Bellman optimality operator. Line 22: 'In this case, the learning procedure interleaves the policy improvement operator with a projection operator, which finds the best approximation of this improved value function in the space of realizable value functions.' Policy improvement is wrong and only in certain algorithms (such as TDC/GTD2 etc) is a projection operator explicitly used. I think what you mean is that the TD fixed point is the solution to the projection of the Bellman operator into the function space, PTV=V? References should really be given here, as it would help to work out what you are actually claiming. Line 28: 'Although this property can be used to show the convergence of such methods in the deterministic setting, it does little to our understanding of the relationship of these methods to value-based ones.' If one has access to true unbiased estimates and standard rm conditions are satisfied, stochastic gradient descent will converge. This doesn't involve deterministic updates. There have been several papers proving convergence when stochastic updates are used in actor-critic settings which use both value-based updates and policy improvement e.g. [1] [2] Line 67: 'The parameter updates in Eq. 2 and Eq. 4 involve the current current policy πθt twice' No, fours times. One in dπt(s), one in πt(a|s), one in Qπt(s,a) and one in the log term. [1] Actor-critic Algorithms, Konda et al. 2000 [2] Convergent Reinforcement Learning with Function Approximation: A Bilevel Optimization Perspective, Yang et al 2018

Clarity: The clarity of this paper is very poor. Value-based is used to mean different things at different points in the paper. This makes the paper very confusing. There is value based to mean value based methods such as Q-learning or SARSA (although no reference to these algorithms or anything like them is made) in the introduction and then value-based to refer to the policy gradient theorem presented in [3]. As discussed in the correctness section, much of the paper is ambiguous and seems wrong or confused in its claims. This is further compounded by the lack of references, so it is very difficult to understand what the authors are claiming. Propositions 2 and 4 are trivial one line results that arise from the fact that the derivative of the RL objective is zero for an optimal policy. I don't see the need for a whole separate theorem in the Appendix when this could simply be written in the main text. I find the paragraph starting line 79 very confusing and can't work out what it means I find the paragraph starting line 158 very confusing and can't work out what it means, especially the line 'Although this result is of no practical interest since it requires knowing π(θ∗) in advance, it proves that there are better sampling distributions than the current policy and that, in some sense, off-policy learning without importance correction is “optimal” when sampling from the optimal policy.' I've worked extensively in the RL as inference field and have no idea what the sentence beginning line 267 : 'Unlike the operator view, the control-as-inference formulation is limited because rewards often cannot easily be interpreted as probabilities, making it also difficult to establish connections with RL algorithms outside the formulation like PPO.' means What's decided as traditional or classical seems arbitrary; I wouldn't call MPO a traditional RL algorithm for example. There are lots of references to words like 'bigger', 'better' and 'good' with no precise qualification or meaning. This makes the writing style very unscientific in some instances. The use of pi for both a policy or trajectory is incredibly confusing throughout. There is little coherence between the sections, they just seem to be disjointed ideas or bits of information. Section 4 for example looks at changing the rewards, introducing new divergences, an empirical evaluation halfway through, changing the rewards again, introducing a new operator that recovers PPO and introducing a new operator that recovers MPO. There is no narrative connecting these sections and subsections. Some of the sections are confusingly named, for example, subsection 4.1 is labelled 'moving beyond rewards' yet all the resulting subsubsections (4.1.1 and 4.1.2) are about changing the rewards, although in the subsection 4.1.1 named 'polynomial rewards', there is no reference to polynomial rewards except in the figure caption of an empirical evaluation. [3] Policy Gradient Methods for Reinforcement Learning with Function Approximation, Sutton et al 2000

Relation to Prior Work: There are very few precious references to prior works and it is not clear that what is being presented here offers any insights beyond existing work. Again, this may be down to the issues of clarity.

Reproducibility: Yes

Additional Feedback: This paper was really difficult to read. Whilst the maths is straightforward, the writing style, use of overloaded notation, imprecise language, illogical structure and fundamental errors make it difficult to find any motivation in the work. I've tried reading it several times now, but can't spend any longer attempting to piece it together. POST-REBUTTAL FEEDBACK: I don't believe that my concerns have been addressed by the rebuttal. Following discussion I believe that I understand the contribution of the paper, but the clarity of work really does let it down. To summarise: USE OF VALUE-BASED: Line 19: 'In particular, value-based methods are often cast as the iterative application of a policy improvement operator, the Bellman optimality operator, which transforms the value function into a “better” one' This only Q-learning applies the Bellman optimality operator. SARSA is a value based method that doesn't do this. Please make references to the specific algorithms you mean here. Line 60: '2.2 Value-Based Formulation' Now value based means something entirely different, please clarify this. CONVERGENCE OF POLICY GRADIENT METHODS: Line 27: 'Although this property can be used to show the convergence of such methods in the deterministic setting, it does little to our understanding of the relationship of these methods to value-based ones.' Policy gradients can be shown to be convergent under stochastic gradient descent, that's a major benefit they have over value based approaches. What do you mean by convergent here? Please clarify this and give references or qualifying explanations. What does this mean in reference to line 46 that 'In this paper, we are not interested in the stochastic gradient updates, but the updates on expectation. Thus, our presentation and analyses use the true gradient of the functions of interest.' POLYNOMIAL REWARDS: Line 185 Iτk : π −→ R^kπ I'm really confused about the notation here. Assuming R^k means applying the operator k times, would imply that the return is raised to the k, right? This seems a really strange thing to do, and I can't understand how it is related to polynomial rewards, only polynomial returns. It's therefore very difficult to understand the benefit of this. This is further compounded by the naming of the entire section as 'moving beyond rewards'. Please consider how this section is named and presented. EXPONENTIAL REWARDS AND CONTROL AS INFERENCE: 'The most common such transformation is the exponential function, used by PPO and MPO, leading 􏰄to the operator $\mathcal{I}_{\tau}^{\exp,T}(\pi)=\exp(\frac{R}{T})\pi' is strange as PPO and MPO both introduce a one-step penalty in their objectives which does not result from exponentiated returns (MPO starts this way, but to derive a practical algorithm, an approximation is used to make the KL penalty one step). Using exponentiated returns results in an EXPECTED penalty, leading to, for example with an expected entropy bonus, a soft Q-function. Again, I think I know what the authors are trying to say here, that there are algorithms derived from a principal of exponential reward to ensure positivity, but in isolation this is a weird point to make as it offers little insight and is misleading because these algorithms cannot be derived the actual operator presented. This is made even more confusing to pick apart as the introduction to this section gives the motivation 'we recover operators that give rise to PPO [16] and MPO [1]. I agree that the operator in 4.3 clearly gives rise to PPO, but why reference this algorithm in relation to an operator that doesn't in an early section? Please consider clarifying this section. On line 267: 'Unlike the operator view, the control-as-inference formulation is limited because rewards often cannot easily be interpreted as probabilities, making it also difficult to establish connections with RL algorithms outside the formulation like PPO.' There's plentiful work discussing how an optimality variable can be introduced that is proportional to reward, which defines a probabilisitic graphical model on which inference of a posterior or the maximum likelihood parameters can be carried out, hence this isn't a valid criticism of the RL as inference framework. Maybe what the authors mean here is that the semantics of these optimality variables are somewhat controversial in certain contexts, but again, there are no references here to any RL as inference works that explore these relationships. Please reference this work to clarify what you mean. It's a shame because I think the point being made here is that the operator view is more general than the RL as inference framework, which is a much stronger argument and gives clear motivation for the proposed work. Small error: Line 40: 'p:S×A→S' the transition distribution maps to a space of probability measures on S, not S.


Review 4

Summary and Contributions: This paper gives a new view on the policy gradient methods based on two operators: Policy improvement operator and projection operator. The authors interpret several properties of policy gradient methods using the two operators. Finally, they provide a bridge between value-based methods and policy-based methods by interpolating two types of method by changing a parameter \alpha.

Strengths: This paper is a theoretical paper which gives an interesting view on the policy gradient methods. The authors show that this view can help the readers understand the policy gradient methods by giving an interpretation of properties of policy gradient methods using the proposed operators. The most interesting part in this paper is that this new view can interpolate value-based methods and policy-based methods though a parameter \alpha. Although there are several works bridging the two methods, these works focused on equivalence between an entropy regularized version of them. However, the proposed view can directly interpolate between two types of methods without any regularization term.

Weaknesses: Although this paper gives a different view on the policy gradient methods to readers, there are some limitations: 1. The theory used in its propositions are not in depth as compared to other theoretical papers in this area. The policy improvement operator and the projection operator can be found in few lines, and all propositions except for the proposed surrogate loss and the bridge between value-based and policy-based methods, are also followed by few lines. 2. It seems that the proposed view can be used for interpretation and understanding of RL algorithms, but not designing new practical RL algorithms. Since the policy improvement operator maps a parameterized policy to a better policy that may not be included in the set of parameterized policies, the result of improvement operator cannot be implemented in a practical manner.

Correctness: I followed all the proofs of all propositions and they seem to be correct. In Fig 1, the authors compared the loss function of CPI and the proposed surrogate loss function, and they mentioned that the loss function of CPI is linear approximation of the true loss function. However, this is not correct. CPI also uses a surrogate function which is linear approximation term plus constant times a kind of KL divergence. TRPO and PPO use some modification of this surrogate function to update their policy parameters, but the fundamental loss function is a surrogate function proposed in CPI. Therefore Fig 1 and its explanation are not correct.

Clarity: This paper is well organized and is clearly written.

Relation to Prior Work: This paper discussed the difference in contributions of this work and previous related works.

Reproducibility: Yes

Additional Feedback: ====== After author response ====== I thank the authors response. I've read the response of my concerns and most of them are solved. However, there are some remaining issues. 1. The theoretical results are not in depth relative to other theoretical papers. However, this can be improved by giving additional results of $I^\alpha_V \circ P^\alpha_V$ for different alphas in [0, 1] on more hard environments, like MuJoCo, and, thus, showing the possibility for designing a good practical algorithm. 2. Also, I agree with R3 that this paper abuses or wrongly uses some terminologies and misses links between sections. I think that this paper has a good novelty, but I do not change my score because this paper could be further improved by addressing these issues.

[Author Response · NeurIPS 2020]

We thank the reviewers for the comments and constructive feedback and we are delighted that they appreciated the clarity of the paper and the novelty of the approach. Reviewer 2 lamented the lack of experiments and Reviewer 4 the lack of new practical algorithms. While we wholeheartedly agree that these would be great additions, and we are currently working on both, we believe this current set of results to be of enough interest to the community to develop their own practical algorithms. Please find responses to specific comments below; we will update the paper to reflect this discussion and to fix all other minor points.

**Reviewer 1**

*"In Section 2, it is assumed that the state and action spaces are finite ..."* We assume finite state-action spaces for simplicity in presentation of the proofs, but the results extend to continuous state-action spaces. The deterministic transition model is a typo, and we will update it appropriately.

*"Proposition 6: I am a little confused about the notation ..."* This is a typo in Proposition 6: the remark should use the state-action improvement operator. An analogous statement does apply for the trajectory formulation (Proposition 10).

*"Proposition 5: ... Are there some sufficient conditions to enforce that $Var(R) > 0$ along the whole learning process? ..."* $Var(R) > 0$ can typically be enforced by exploration strategies, which are outside the scope of this paper. However, even without these strategies, REINFORCE is guaranteed to converge to a (potentially suboptimal) stationary point using results from optimization theory.

*"Proposition 2: ... There can be other fixed points?"* There can indeed be other fixed points of the operator: these corresponds to suboptimal stationary points of the expected reward objective $J(\pi)$. Under certain conditions, e.g. tabular policy classes (Agarwal et al 2019), the optimal solution is known to be the only fixed point, but characterizing all fixed-points under arbitrary function approximation remains an open challenge.

**Reviewer 2**

We will organize the related work to be more clear, and add discussion about AlphaGo-Zero in the updated manuscript.

*"It would be interesting to more empirically analyze the induced interpolated algorithm ... on more realistic benchmarks*

We agree that this would be interesting to analyze on more challenging domains, and in fact, we have several such experiments in mind. In this submission, we decided to focus on the theoretical aspects of the work and presentation of the main ideas – a more empirical work is to follow.

**Reviewer 3**

We are sorry to hear that you found the paper to be unclear. We appreciate your specific comments about our introduction section, and will update the paper to improve the precision of these statements and better organize the related work.

We emphasize that we are not proposing a new objective, but rather providing a new way to interpret the original policy gradient objective using ideas of policy improvement. While our work is related to the pseudo-likelihood in RL-as-inference (see discussion in Section 4.3), they differ crucially in how the objective is interpreted. Whereas RL-as-inference interprets the RL objective as inference in a graphical model, our paper re-interprets the objective as application of an improvement and a projection step. This alternative framework provides new insights about the behavior of existing policy gradient methods (e.g. Prop 5, 6) and enables the potential for new algorithms (e.g. Prop 8).

**Reviewer 4**

*"... the result of improvement operator cannot be implemented in a practical manner."* We apologize for the misunderstanding here – a practical algorithm can in fact be created using an improvement operator that produces policies not realizable in our function class. Fundamentally, this is because we never need to explicitly represent the improved policy $\mathcal{I}(\pi)$, only the *projected* improved policy $\mathcal{P} \circ \mathcal{I}(\pi)$.

For a practical algorithm, this requirement primarily boils down to the ability to *implicitly* represent the improved policy (e.g. by weighting previously seen state-action pairs by rewards). In fact, all the existing policy gradient methods we study in this paper (REINFORCE, PPO, and MPO) use weighted implicit policies to implement improvement operators that can produce potentially unrealizable policies. We will update the manuscript to reflect this discussion.

*"... CPI also uses a surrogate function which is linear approximation term plus constant times a kind of KL divergence."*

We apologize for the confusion; we will update Figure 1 with the trust-region penalty typically used with CPI. Please note though that the KL terms in the two approximations serve different purposes: in our bound, the KL term promotes closeness to the *improved* policy, whereas in CPI, it penalizes distance to the *original* policy.

[Meta-Review · NeurIPS 2020]

3 referees advocate to accept the paper due to its novelty and theoretical contribution to understand policy gradient methods. 1 referee (R3) has concerns with ambiguity in the introduction and overstatements of the results. I agree that the writing and ambiguity of some statements need to be improved. A few of these concerns have been addressed by the rebuttal, but could still not convince R3. However, as the theoretical contributions of this paper are significant and the paper shows interesting connections of well known algorithms, I still advocate acceptance. The authors however have to take the comments from R3 into account to improve clarity: - clearly specify to which algorithms you are referring to (e.g. for value-based algorithms, most of your statements are true for Q-learning but not for Sarsa) - In Section 4.1 you show that using an exponential transform results in a fixed point of the operator and claim that this directly relates to MPO and PPO. However, the operators for MPO and PPO are slightly different and it is unclear if the results also holds for these operators. Please discuss this in more detail. - Mention that the entropy regularization of the PPO algorithm is a variant of PPO, that is not used in all papers. For example, in the original paper, no entropy regularization is used for the mujoco experiments. Please also discuss how the operator looks like without entropy regularization.